# Enzymatic Preparation, Identification by Transmembrane Channel-like 4 (TMC4) Protein, and Bioinformatics Analysis of New Salty Peptides from Soybean Protein Isolate

**DOI:** 10.3390/foods13172798

**Published:** 2024-09-03

**Authors:** Ziying Zhao, Mingzhe Yang, Zhijiang Li, Huacheng Tang, Xuejian Song, Xinhui Wang

**Affiliations:** 1College of Food Science, Heilongjiang Bayi Agricultural University, Daqing 163319, China; czzyc27999@163.com (Z.Z.); woshiyangerlv@163.com (M.Y.); byndsxj@126.com (X.S.); w604466213@163.com (X.W.); 2National Coarse Cereals Engineering Research Center, Daqing 163319, China; 3Heilongjiang Food and Biotechnology Innovation Research Center, Daqing 163319, China

**Keywords:** soybean protein isolate, saltiness peptides, salty receptor protein TMC4, molecular docking

## Abstract

To address the public health challenges posed by high-salt diets, this study utilized pepsin and flavourzyme for the continuous enzymatic hydrolysis of a soy protein isolate (SPI). The separation, purification, and identification of salt-containing peptides in SPI hydrolysate were conducted using ultrafiltration (UF), gel filtration chromatography (GFC), and Liquid Chromatography–Mass Spectrometry/Mass Spectrometry (LC-MS/MS). Subsequently, a molecular docking model was constructed between salt receptor protein transmembrane channel 4 (TMC4) and the identified peptides. Basic bioinformatics screening was performed to obtain non-toxic, non-allergenic, and stable salt peptides. After the enzymatic hydrolysis, separation, and purification of SPI, a component with a sensory evaluation score of 7 and an electronic tongue score of 10.36 was obtained. LC-MS/MS sequencing identified a total of 1697 peptides in the above component, including 84 potential salt-containing peptides. A molecular docking analysis identified seven peptides (FPPP, GGPW, IPHF, IPKF, IPRR, LPRR, and LPHF) with a strong theoretical salty taste. Furthermore, residues Glu531, Asp491, Val495, Ala401, and Phe405 of the peptides bound to the TMC4 receptor through hydrogen bonds, hydrophobic interactions, and electrostatic interactions, thereby imparting a significant salty taste. A basic bioinformatics analysis further revealed that IPHF, LPHF, GGPW, and IPKF were non-toxic, non-allergenic, and stable salt-containing peptides. This study not only provides a new sodium reduction strategy for the food industry, but also opens up new avenues for improving the public’s healthy eating habits.

## 1. Introduction

Salty taste, often regarded as the “king of flavors”, plays a fundamental role in our diet and is the only taste with significant physiological effects; it regulates the osmotic balance between cells and blood and normal water and salt metabolism. However, high-salt diets are strongly associated with hypertension and other cardiovascular diseases [1]. The World Health Organization (WHO) recommends a daily salt intake of no more than 5 g, but Chinese Dietary Guidelines indicate that about 80% of people exceed this recommendation. However, the addition of salt makes an important contribution to the flavor and edible quality of many foods (e.g., breads, processed soy products, potato chips, biscuits, canned products). In response to increasing consumer demand for healthier food options, the food industry will make more far-reaching efforts to reduce salt. Therefore, researchers have been exploring various strategies to reduce salt intake while maintaining desirable flavor. One promising approach is the development of salt substitutes that ensure a salty taste, to reduce sodium ion intake, and offer health benefits. And many researchers have attempted various methods to salt reduction, including optimizing the structure and distribution of salt [2], and using metal chlorides [3] and flavor enhancers to reduce sodium salt intake [4], with the most attention being focused on the study of salty peptides such as Orn-β-Ala·HCl and Orn-Tau·HCl [5]. These peptides do not contain Na^+^ but have a salty taste comparable to or even stronger than sodium chloride, which has inspired further research into their taste mechanisms and applications. Salty peptides, as novel food additives, have shown potential for replacing table salt without sacrificing flavor, gaining significant attention in recent years [6].

Salty peptides can be derived from various sources, including animal [7,8,9], microbial [10,11,12], and plant sources. Among these, plant-derived salty peptides are particularly noteworthy, with legumes being a rich source. Legumes, due to their high protein content, and richness in salty and umami amino acids, provide an excellent basis for extracting salty peptides. For instance, enhanced salty effects were achieved by deep enzymatic hydrolysis of pea protein [13]; salty peptide EDEGEQPRPF was isolated from commercial soy milk, demonstrating its salty taste enhancement equivalent to 63 mmol/L NaCl [14]. Additionally, several salty dipeptides such as Ile-Gln, Pro-Lys, Ile-Glu, Thr-Phe, and Leu-Gln have been identified in soy sauce [15]. Despite these advancements, there is still a scarcity of research focused on the enzymatic hydrolysis of a soy protein isolate (SPI) to extract salt-containing peptides. This gap presents a significant opportunity for future investigations aimed at exploring SPI for the development of effective sodium salt substitutes.

Transmembrane channel protein 4 (TMC4) is a significant protein recently identified in the field of taste physiology [16]. It is expressed in taste bud cells and plays a crucial role in the transmission of taste signals. TMC4 is highly expressed in the fungiform and circumvallate papillae, which project to the glossopharyngeal nerve and mediate responses to high concentrations of NaCl. Electrophysiological analyses using HEK293T cells have shown that TMC4 functions as a voltage-dependent Cl^−^ channel, with currents completely inhibited by the anion channel blocker NPPB [17]. The action mechanism of salty peptides primarily involves ionizing cations that enter the cell through transient receptor potential vanilloid (TRPV) channels on the cell membrane, leading to calcium ion polarization. This influx of calcium ions triggers the release of neurotransmitters, activating the next level of neurons. Neural signals are then transmitted to central brain regions, such as the insula and orbitofrontal cortex, where they are encoded as taste signals, producing the salty taste sensation [18]. Despite existing research on the effects of various peptide substances on taste [19], there is still less information on the interaction between specific salty peptides and TMC4 [20]. Therefore, studying the molecular interactions between TMC4 and soybean-derived salty peptides is essential for understanding the functional mechanisms of salty peptides and provides a theoretical foundation for developing novel food seasonings.

The aim of this study is to use SPI as a raw material to prepare novel salty peptides through enzymatic hydrolysis, separation, purification, and identification. Sensory evaluation and electronic tongue analysis techniques were employed to characterize these peptides. Furthermore, molecular docking with the TMC4 receptor protein was used to elucidate the taste mechanism, and basic bioinformatics methods were applied to screen for higher-quality salty peptides. This research not only provides new insights into the taste effects of salty peptides but also offers theoretical references for their application in the food industry, thereby promoting public health.

## 2. Materials and Methods

### 2.1. Materials and Chemicals

SPI were obtained from ANYANG BEIJIA FOOD CO., Ltd. (Anyang, Henan, China). Pepsin (30,000 U/g), flavourzyme (30,000 U/g), citric acid, and Sephadex G-10 were obtained from Shanghai Yuanye Bio-Technology Co., Ltd. (Shanghai, China). Food-grade monosodium glutamate (MSG) was purchased from Lianhua Holding Company Limited (Xiangcheng, China). Sodium chloride (NaCl) was purchased from China National Salt Industry Group Co., Ltd. (Beijing, China). Caffeine was purchased from Shandong Jukang Biotechnology Co., Ltd. (Jinan, China). HCl was purchased from Liaoning Quanrui Reagent Co., Ltd. (Jinzhou, China). The purity of the above chemicals is analytically pure.

### 2.2. Preparation of SPI Enzymatic Hydrolysate

The enzymatic hydrolysis of SPI was performed as reported with slight modifications [21]. SPI was sequentially hydrolyzed by pepsin and flavourzyme. The SPI was dissolved in 98 mL of deionized water at a substrate concentration of 2% (*w*/*w*) and adjusted to pH 2.0 with 2 mol/L HCl. Pepsin was then added to the SPI, and the mixture was incubated at 37 °C for 6 h with an enzyme-to-SPI ratio of 4%. Subsequently, the reaction temperature was raised to 50 °C, and the pH was adjusted to 7.0 with 3 mol/L NaOH. Flavourzyme was added, and the protein was further incubated for 4 h with an enzyme-to-SPI ratio of 1.5%. After enzymatic hydrolysis, the enzyme was inactivated by heating in a water bath at 95 °C for 20 min. The SPI enzymatic hydrolysate (SPIEH) was then centrifuged at 4000 rpm for 15 min, freeze-dried, and stored at −20 °C for a further analysis.

### 2.3. Molecular Weight Distribution of SPIEH

The SPIEH was determined using an integrated gel permeation chromatography (GPC) system (LC20, Shimadzu, Jingdu, Japan). SPIEH (5 mg/mL) was absorbed with a syringe and filtered with a 0.45 μm molecular membrane. A 100 μL filtrate was absorbed by a microsampler and injected into the GPC column.

### 2.4. Determination of Free Amino Acid Content in SPIEH

The free amino acid contents were determined using a Biochrom30+ amino acid analyzer (Biochrom Ltd., Cambridge, UK) following the reported method with some modifications [22]. A 1 mL aliquot of the supernatant was mixed with 1 mL of 5% sulfosalicylic acid. The separation was performed using a Na-type cation resin chromatography column (200 mm × 4.6 mm, 3 µm particles) with UV detection at a wavelength of 440 nm. The column temperature was programmed to increase from 55 °C to 65 °C to 77 °C, and the injection volume was 20 µL with a flow rate of 10 mL/h.

### 2.5. Evaluation of Salt Taste

#### 2.5.1. Sensory Evaluation

The sensory evaluation method was adapted with slight modifications [23]. The sensory panel consisted of 14 trained individuals (7 males and 7 females, ages 22 to 32, with no history of abnormal taste perception). Following sensory training [14], evaluations were conducted in a controlled room at 25 ± 1 °C with no communication allowed between participants. The lyophilized powders of enzymolysis from different fractions obtained from multiple collections were prepared into 1% (*w*/*v*) solutions. Aqueous solutions of 0.08% citric acid, 0.08% caffeine, 0.35% NaCl, and 0.35% MSG were served as the standard references for sour, bitter, salty, and umami tastes, respectively. The taste characteristics of each solution were assessed, and the intensity of the salty taste was rated on a scale from 0 to 10, where 0 indicated no taste and 10 indicated the strongest taste [24].

#### 2.5.2. Electronic Tongue Analysis

The sample solutions of the same concentration were prepared using a 600 mg/L NaCl solution as the control. The prepared solutions were then poured into the special measuring cup of an electronic tongue (ALPHA MOS (Shanghai) Instrument Trading Co., Ltd., Shanghai, China) and measured at room temperature. Each sample was measured in triplicate, with the instrument stabilizing after the first measurement. The average value of the last three signal data points was used as the sample’s taste signal intensity.

### 2.6. Separation and Purification Salty Peptides from SPIEH

#### 2.6.1. Separation by UF

Three hundred grams of SPIEH was dissolved in 15 L of ultrapure water and completely dissolved using a KQ 5200 V ultrasonic cleaner (Ningbo Xinzhi Biotechnology Co., Ltd., Ningbo, China). A UF system (Jilin Haipu Technology Development Co., Changchun, China) was used to separate proteins based on molecular weight. The SPIEH solution was first passed through a 10 kDa UF membrane, followed by a 5 kDa UF membrane, and then desalted using a desalting column. Further separation was performed using 3 kDa and 1 kDa UF membranes (Millipore Corporation, Shanghai, China). After multiple filtration steps, three peptide sample solutions with molecular weights of 3–5 kDa, 1–3 kDa, and <1 kDa were obtained. These samples were collected, and freeze-dried after spin evaporation, and the group with the highest salt content was selected for further separation and purification based on the evaluation of salt taste.

#### 2.6.2. Purification by GFC

The UF group (MW < 1 kDa) was selected for GFC. The freeze-dried samples were dissolved in ultrapure water to a concentration of 50 mg/mL and then passed through a 0.45 μm filtration membrane. The sample solution was separated using an AKTA protein purifier (GE Healthcare, Chicago, IL, USA) equipped with a Sephadex G-10 gel chromatography column (2.6 × 80 cm, Smart-Lifesciences Corp., Changzhou, China). A 1 mL aliquot of the filtered solution was injected into the gel column, and separation was performed using ultrapure water as the eluent at a flow rate of 0.60 mL/min, with UV detection set to 220 nm. The fractions were collected from multiple injections, concentrated, and freeze-dried for the evaluation of salt taste.

### 2.7. Identification of the Peptide Sequence

The sample from GFC was desalted using ZipTip C18, vacuum-dried, and analyzed by LC-MS/MS (Thermo, Waltham, MA, USA). The chromatographic separation was performed on a PepMap RSLC C18 (75 μm × 150 mm, 2 μm, 100 A) column with 0.1% formic acid in water as mobile phase A and 0.1% formic acid in acetonitrile (ACN) as mobile phase B. A 4 µL sample was injected, with UV detection at 214 nm and a flow rate of 0.25 μL/min. The elution gradient was as follows: 0–10% B from 0 to 8 min, 10–15% B from 8 to 33 min, 15–28% B from 33 to 43 min, 28–40% B from 43 to 50 min, 40–95% B from 50 to 65 min, and 95–5% B from 65 to 70 min.

Electrospray ionization (ESI) was used as the ionization source for a Mass Spectrometry analysis. The mass spectrum was scanned in the *m*/*z* range of 300–1400. Nitrogen was used as the collision gas with a flow rate of 250 μL/min. The resolution of the primary mass spectrum was 7 × 10^4^ (for *m*/*z* ≤ 200), with an automatic gain control (AGC) target set to 5 × 10^4^, and the collision energy for the secondary mass spectrum was set between 12 and 14 eV. The Mass Spectrometry error tolerance was set to 0.05 Da, and the resolution for the secondary mass spectrum was 1.75 × 10^4^ (for *m*/*z* ≤ 200). A data analysis was performed using the PEAKS 12.0 software with the Uniprot proteome database (www.uniprot.org; proteome UP000008827_20240125.fasta) for peptide identification and a semi-quantitative analysis.

### 2.8. Construction of Saltiness Receptor Model TMC4

The TMC4 (transmembrane channel 4-like) protein amino acid sequence was obtained from the NCBI database (https://www.ncbi.nlm.nih.gov/, accessed on 23 April 2024; accession number: NP_001138775). Initial modeling using SWISS-MODEL (https://swissmodel.expasy.org/, accessed on 23 April 2024) indicated that the homologous protein model had a low consistency of 22.94%, which is below the 30% threshold for reliable models. Consequently, the TMC4 protein model was created from scratch using the deep learning model AlphaFold2, and the model with the highest predicted accuracy was selected. The overall quality of the model was assessed using the Ramachandran plot and the ERRAT method (https://services.mbi.ucla.edu/errat/) (accessed on 23 April 2024), which evaluated the rationality and reliability of the protein structure.

### 2.9. Molecular Docking of Potential Peptides with TMC4

The 3D structure of the polypeptide was constructed using Discover Studio and saved as a PDB file with minimal energy configuration. The receptor protein was prepared by adding polar hydrogen atoms and assigning equilibrium charges using AutoDockTools 1.5.6 software. Both the receptor protein and the ligand molecules were converted into PDBQT format for molecular docking. AutoDock Vina 1.1.2 was employed to perform global molecular docking simulations between the receptor protein and the ligand molecules. The docking results were analyzed visually using PyMOL 3.0 and Discover Studio 2023. AR, RG, and RS peptides were used as positive controls. The top seven peptides with the most favorable free energy of binding were selected for a detailed visual analysis using Discovery Studio.

### 2.10. In Silico Screening of Premium Salty Peptides

A series of in silico tools were integrated to identify candidate salty peptides with favorable safety profiles. The ToxinPred tool (https://webs.iiitd.edu.in/raghava/toxinpred/index.html) (accessed on 5 May 2024) was used to screen for potential toxicity, AllerTOP (https://www.ddg-pharmfac.net/AllerTOP/) (accessed on 5 May 2024) was employed to assess allergenicity, and Expasy ProtParam (https://web.expasy.org/protparam/) (accessed on 5 May 2024) was utilized to evaluate peptide stability [25].

### 2.11. Statistical Analysis

All the data were analyzed using IBM SPSS Statistics 27.0 software (SPSS Inc., Chicago, IL, USA), and the experiment was repeated in triplicate. The chart was prepared using Origin 2024 (OriginLab Inc., Northampton, MA, USA).

## 3. Results and Discussion

### 3.1. Molecular Weight Distribution of the Hydrolysate

The distribution percentages of peptides with different molecular weights in SPIEH are shown in Table 1. Peptides are intermediated products of protein hydrolysis, and their molecular weights are generally less than 10 kDa. In SPIEH, the polypeptide distribution is mainly concentrated in the 100 Da–1 kDa range, accounting for nearly 50%, which is significantly higher than those of peptides in the 3–5 kDa and 1–3 kDa ranges. Saltiness and saltiness-enhancing peptides may be related to the low molecular weight (<1 kDa) of the peptides [26]. This distribution is consistent with the results of previous studies [9]. It can be inferred that the dual-enzyme continuous enzymatic hydrolysis effectively achieved the desired outcome, resulting in a higher content of small-molecular-weight peptides.

### 3.2. Free Amino Acid Content Analysis

The composition and content of free amino acids have a significant influence on taste [27]. A previous study has found that peptides containing more hydrophilic amino acids (e.g., Ser, Glu, Thr, Gly) tend to exhibit a stronger salty taste [28]. Additionally, polypeptides with umami amino acids can also possess a salty taste [29]. In this study, a total of 16 amino acids were detected in SPIEH (Table 2), with hydrophilic amino acids making up 51.87% of the total content and umami amino acids constituting 38%. These proportions suggested that the sample may contain salty peptides or salt-enhancing peptides.

### 3.3. Separation and Purification of SPIEH

The components of SPIEH with molecular weights of 3–5 kDa, 1–3 kDa, and <1 kDa were obtained through UF separation, respectively. According to the results of the salt taste evaluation system (electronic tongue analysis (Figure 1A), sensory evaluation (Figure 1B)), the peptide with a molecular weight of <1 kDa exhibited the strongest saltiness, with a sensory evaluation score of 9 and an electronic tongue score of 10.77. The peptide with a molecular weight of 1–3 kDa displayed moderate saltiness, with a sensory evaluation score of 7 and an electronic tongue score of 9.74. Conversely, the peptide with a molecular weight of 3–5 kDa had the weakest saltiness, with a sensory evaluation score of 3 and an electronic tongue score of 5.17. The high correlation between the electronic tongue results and sensory evaluation scores indicates that saltiness intensity increases as molecular weight decreases. These findings are consistent with those of Zheng et al. [12], who reported that peptides with a molecular weight of <1 kDa in yeast extracts exhibited strong umami and salty flavors. The saltiness of a peptide usually refers to the electrolyte content or charge effect of the peptide in a solution [30]. The reason why smaller-molecular-weight peptides may exhibit higher saltiness may be because smaller-molecular-weight peptides have an increased charge density, and their stronger water solubility makes them easier to dissociate. At the same time, the higher diffusion rate makes it easier for their charged groups to exchange with ions in a solution, thus increasing the salinity. Thus, the <1 kDa peptides after UF were selected for further separation and purification.

The <1 kDa peptides obtained through UF were subsequently separated and purified using GFC, yielding the components G1, G2, G3, and G4 (Figure 1C). The electronic tongue analysis (Figure 1D) and sensory evaluation (Figure 1E) revealed that all components exhibited saltiness, with the G2 component demonstrating the strongest saltiness (sensory score of 9 and electronic tongue score of 10.36), while the other three components exhibited more pronounced bitterness. Therefore, G2 was selected for further experiments, including distillation, freeze-drying, and storage at −20 °C for subsequent amino acid sequence identification.

### 3.4. Peptide Sequence Identification

In this study, LC-MS/MS was employed to determine the peptide sequences of the G-2 component, and database searches using PEAKS 12.0 software identified a total of 1697 peptide sequences, most of which consist of 4–6 amino acids, with 712 being tetrapeptides. Then, 84 possible savory peptides were selected for molecular docking [31] with TMC4 receptor protein by amino acid types, peptide chain length, and peptide score (Table 3). Generally, shorter peptide sequences exhibit stronger salty tastes [32]. Additionally, amino acids located at or near the C-terminus with positively charged R groups, such as Arg (R) and Lys (K), play a significant role in the salty taste of peptides [14]. The role of umami amino acids, such as Glu (E) and Asp (D), should also not be overlooked when searching for new salty peptides [33]. There is a positive correlation between the perception of salty and umami tastes, and salty peptides can also possess umami flavor. Therefore, 84 peptide segments rich in hydrophilic and umami amino acids with high peptide segment scores were selected for further molecular docking.

### 3.5. Construction and Evaluation of Salty Receptor TMC4

Due to the high sensitivity of the salt taste receptor TMC4 to low-concentration salt solutions [17], transmembrane channel-like protein 4 (TMC4) was chosen as the receptor for this study. Using the deep protein learning model AlphaFold2, the TMC4 protein model was constructed de novo, resulting in five predicted models. As shown in Table 4, the rank1 protein model predicted the local distance difference test (pLDDT) (78.3) and predicted the TM score (PTM) (0.773). Therefore, we chose the rank1 model for subsequent protein model evaluation (Figure 2A). The Ramachandran plot (Figure 2B) shows that only 1.1% of amino acids in the rank1 protein model are in the disallowed regions, while 98.9% are in allowed regions, indicating that the rank1 protein model is reasonable. Using the ERRAT method to evaluate the overall quality of the rank1 protein model (Figure 2C), the overall quality factor score of the rank1 model was 93.604, indicating that the computational error of 93.604% of residues in this model was below the 95% error threshold. Therefore, the results of both evaluation indicators suggest that the rank1 model structure is reasonable, with high credibility, and can be used as a protein receptor for molecular docking.

### 3.6. Molecular Docking of the Identified Peptide to the Salty Receptor TMC4

In this study, we selected AR, RG, and RS [8], known potent and widely sourced salt-containing peptides, as positive controls to provide references for subsequent selection. The 84 tetrapeptides screened above were sequentially docked with the salty receptor TMC4; seven possible salty peptides with the lowest binding energies were obtained from the docking (Figure 3), with a binding energy order as follows: FPPP (−9.4 kcal/mol) < IPHF (−9.1 kcal/mol) = LPHF (−9.1 kcal/mol) < GGPW (−9.0 kcal/mol) < IPRR (−8.8 kcal/mol) < IPKF (−8.6 kcal/mol) = LPRR (−8.6 kcal/mol) < AR (−6.7 kcal/mol) < RG (−6.3 kcal/mol) = RS (−6.3 kcal/mol). Molecular docking technology uses computer simulation software to place ligand molecules at the active sites of receptors. Based on the principles of geometric and energy complementarity, it continuously adjusts parameters such as the positions and conformations of ligands and receptors [34], and uses scoring functions to screen for the best binding mode [35]. It is mainly used to predict the binding mode and interactions between ligands and receptors. Common docking methods include rigid docking, flexible docking, and semi-flexible docking [36]. This study used semi-flexible docking technology. The lower the binding energy of the docking, the more stable the ligand–receptor binding. Comparison with positive controls shows that the binding energies of the seven salty peptides are all lower than those of control peptides (AR, RG, RS), suggesting that their saltiness may be stronger. Further synthesis verification is needed to determine their exact saltiness values.

### 3.7. Molecular Docking of the Identified Peptide to the Salty Receptor TMC4 and Bioinformatics Analysis

The 2D and 3D docking diagrams of the seven peptides with TMC4 are shown in Figure 4. The analysis of the binding sites and binding forces between the peptides and the TMC4 receptor reveals that three main types of molecular interactions are involved: hydrogen bonding, hydrophobic interactions, and electrostatic interactions. Hydrogen bonding and hydrophobic interactions play major roles (Table 5, Figure 5), which is consistent with previous research findings [10,31]. Amino acid residues such as Glu531 and Asp491 are crucial for hydrogen bonding interactions, while residues such as Val495 and Ala401 facilitate hydrophobic interactions with the TMC4 receptor. The electrostatic interactions formed between the peptides and TMC4 primarily involve salt bridges, with GGPW and LPRR forming salt bridges with Glu531, and IPKF, LPHF, and LPRR forming salt bridges with Asp491 and Asp530. Thus, the binding of salty peptides to the TMC4 receptor is mediated through these three types of interactions, resulting in more stable conformations [37]. Additionally, as depicted in Figure 6, Glu531 is the most frequently observed residue, while Asp491, Val495, Ala401, and Phe405 also serve as significant binding sites. Arg151, Thr148, Tyr565, Arg424, Arg330, Tyr677, and Arg580 were identified as key binding sites in the docking of TMC4 with salt-containing peptides from yeast extracts, highlighting the important role of Arg in the binding process [31]. However, Glu plays an important role in the binding of TMC4 to salt-containing peptides in this study. Differences in binding sites observed in our study may be attributed to variations in the templates used for model construction and the specific docking targets employed.

To select higher-quality salty peptides for potential food applications, this study utilized three basic bioinformatics methods to analyze the seven peptides, and the results indicated that IPHF, LPHF, GGPW, and IPKF are non-toxic, non-sensitizing, and stable salty peptides (Table 6).

## 4. Conclusions

In this study, a soybean protein isolate was used as raw material. After hydrolysis by pepsin and flavourzyme, the molecular weight distribution of the hydrolysates was mainly <1 kDa (53%), rich in various hydrophilic amino acids (51.87%) and umami amino acids (38%). The soybean isolate protein hydrolysate (SPIEH) was purified through ultrafiltration (UF) and gel filtration chromatography (GFC), resulting in fractions with a sensory evaluation score of 7 and an electronic tongue score of 10.36. LC-MS/MS identified 84 potential salty peptides. A 3D model of the salty receptor TMC4 was constructed and docked with the potential salty peptides, identifying seven peptides (FPPP, GGPW, IPHF, IPKF, IPRR, LPRR, and LPHF) with lower binding energies. Molecular simulation results indicate that Glu531, Asp491, Val495, Ala401, and Phe405 play key roles in peptide–receptor interactions, contributing significantly to the salty taste of the peptides. The further basic bioinformatics analysis identified IPHF, LPHF, GGPW, and IPKF as four non-toxic, non-sensitizing, and stable salty peptides. In the future, we will study the synthesis of four kinds of salty peptides, and use an electronic tongue to identify the salty intensity.

## Figures and Tables

**Figure 1 foods-13-02798-f001:**
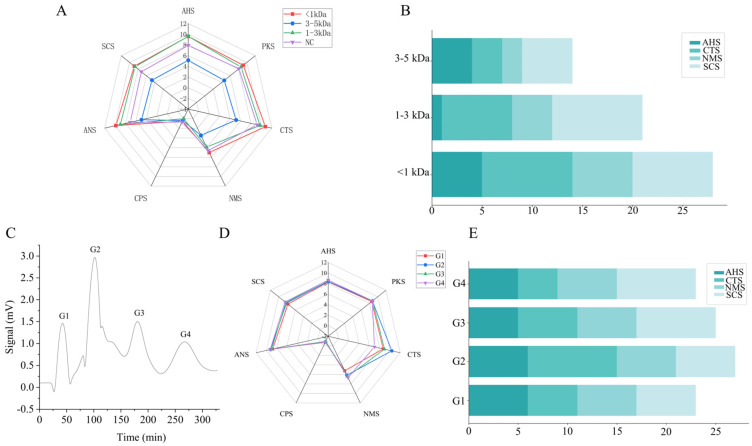
Evaluation of salt taste of UF fractions. (**A**) Electronic tongue score of UF fractions. (**B**) Sensory evaluation score of UF fractions. Separation and purification of GFC fractions and evaluation of salt taste. (**C**) Gel permeation chromatography of MW < 1 kDa fraction. (**D**) Electronic tongue score of GFC fractions. (**E**) Electronic tongue score for GFC fractions. Note: AHS means acid; CTS means salty; NMS means umami; SCS means bitter; ANS means sweet; CPS and CKS are chiasma types.

**Figure 2 foods-13-02798-f002:**
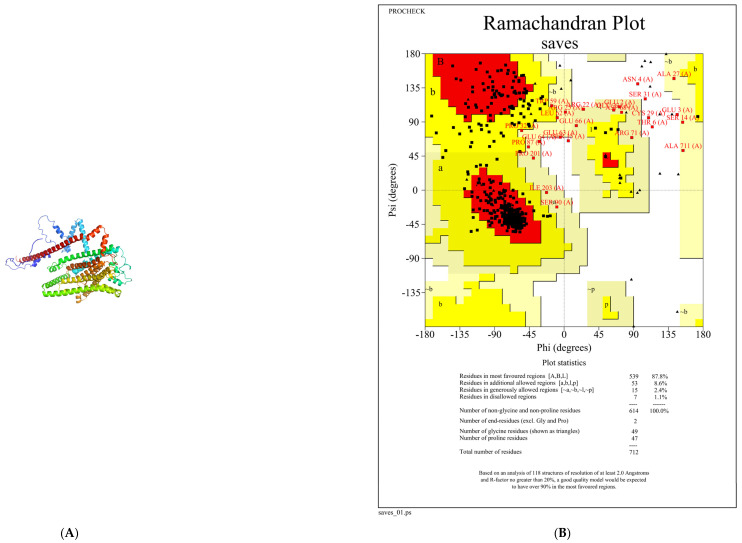
Construction and evaluation of 3D structure of TMC4. (**A**) Three-dimensional image of RANK1 protein model. (**B**) Ramachandran plot of RANK1. (**C**) Error valuation of RANK1.

**Figure 3 foods-13-02798-f003:**
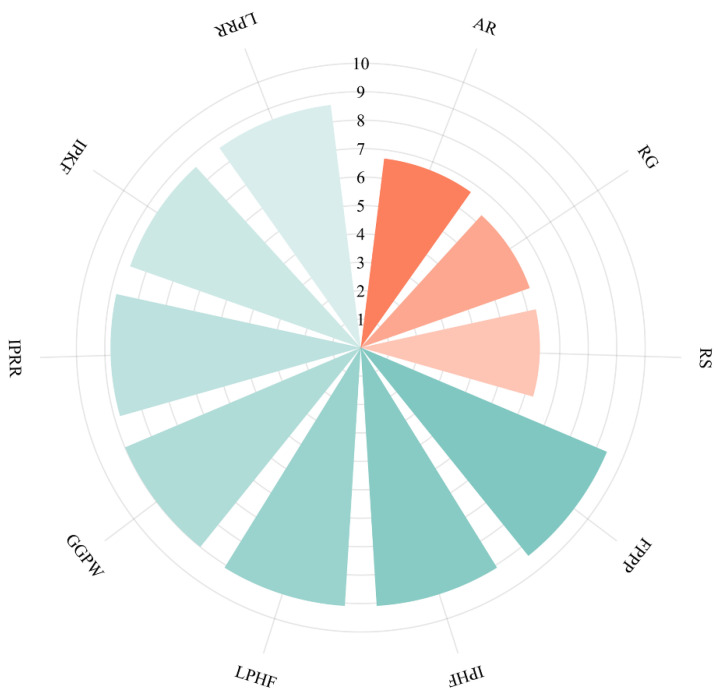
Molecular docking binding energy of 7 selected peptides and positive control peptides.

**Figure 4 foods-13-02798-f004:**
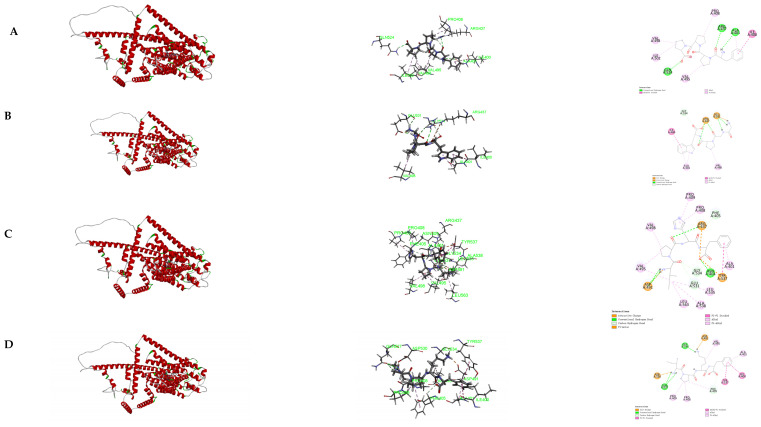
The 3D and 2D structural diagram of docking between salty peptides and TMC4. (**A**) FPPP; (**B**) GGPW; (**C**) IPHF; (**D**) IPKF; (**E**) IPRR; (**F**) LPHF; (**G**) LPRR.

**Figure 5 foods-13-02798-f005:**
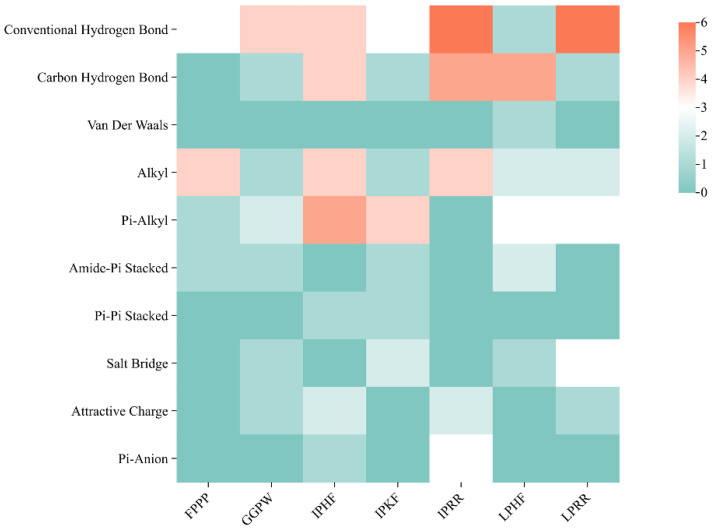
Peptide binding to TMC4 receptor produces interaction type and frequency of interactions.

**Figure 6 foods-13-02798-f006:**
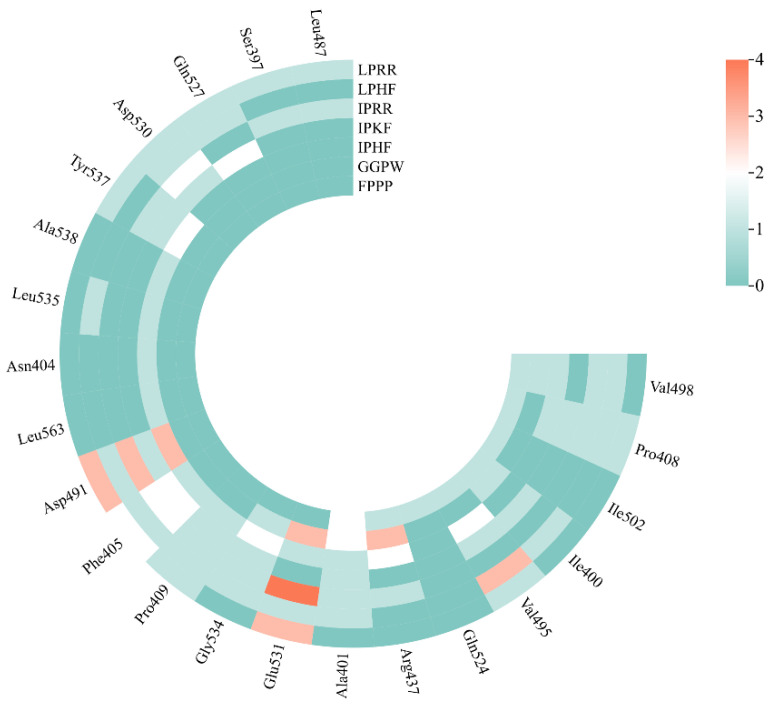
Key amino acid residues around the TMC4 active site.

**Table 1 foods-13-02798-t001:** Molecular weight distribution of SPIEH.

Molecular Weight	Percent (%)
>5 kDa	17
3–5 kDa	10
1–3 kDa	20
100 Da–1 kDa	49
<100 Da	4

**Table 2 foods-13-02798-t002:** Free amino acid contents of SPIEH.

Species	Content (%)
Glu	17.0
Asp	9.49
Leu	7.00
Arg	6.30
Lys	5.20
Ser	4.38
Phe	4.33
Pro	4.10
Ile	3.90
Val	3.83
Ala	3.55
Gly	3.41
Tyr	3.17
Thr	3.10
His	2.14
Met	1.15
Total	82.1

**Table 3 foods-13-02798-t003:** Information on 84 peptide segments.

Peptide Sequence	Peptide Score (−10LgP)	*m*/*z* (kg/C)
KPII	56.97	470.3322
KPIL	56.97	470.3322
KPLI	56.97	470.3322
KPLL	56.97	470.3322
APSI	55.55	387.2217
APSL	55.55	387.2217
GGSW	55.14	406.1700
IGGY	54.74	409.2053
LGGY	54.74	409.2053
SGGW	54.73	406.1700
AASF	54.46	395.1902
GSGW	54.37	406.1700
VGGI	54.25	345.2112
VGGL	54.25	345.2112
IAGY	54.12	423.2215
LAGY	54.12	423.2215
IVGY	54.09	451.2529
LVGY	54.09	451.2529
FGGP	53.85	377.1799
NGVF	53.82	436.2166
IGHI	53.51	439.2643
IGHL	53.51	439.2643
LGHI	53.51	439.2643
LGHL	53.51	439.2643
IPKF	53.43	504.3156
LPKF	53.43	504.3156
GGMI	53.32	377.1833
GGML	53.32	377.1833
GGPW	53.32	416.1907
IPRR	53.17	541.3558
LPRR	53.17	541.3558
AAGY	52.95	381.1743
IYGG	52.89	409.2061
LYGG	52.89	409.2061
PIAP	52.84	397.2425
PLAP	52.84	397.2425
IPAV	52.73	399.2579
LPAV	52.73	399.2579
VIYI	52.71	507.3153
VIYL	52.71	507.3153
VLYI	52.71	507.3153
VLYL	52.71	507.3153
GGSY	52.66	383.1505
TPAF	52.31	435.2211
IPHF	52.29	513.2806
LPHF	52.29	513.2806
ITFI	52.24	493.2996
ITFL	52.24	493.2996
LTFI	52.24	493.2996
LTFL	52.24	493.2996
KGII	52.17	430.3005
KGIL	52.17	430.3005
KGLI	52.17	430.3005
KGLL	52.17	430.3005
AGPY	52.06	407.1905
IAIT	52.03	417.2683
IALT	52.03	417.2683
LAIT	52.03	417.2683
LALT	52.03	417.2683
KIFN	52.00	521.3060
KLFN	52.00	521.3060
GGCF	51.91	383.1327
GGAW	51.78	390.1743
IHIP	51.57	479.2962
IHLP	51.57	479.2962
LHIP	51.57	479.2962
LHLP	51.57	479.2962
KGPR	51.52	457.2856

**Table 4 foods-13-02798-t004:** Protein models predict structural confidence scores and PTM scores.

Model Name	PLDDT	PTM
RANK1	78.3	0.773
RANK2	75.6	0.750
RANK3	75.3	0.743
RANK4	73.1	0.723
RANK5	71.2	0.705

**Table 5 foods-13-02798-t005:** The peptide binds to the TMC4 receptor to produce the type and number of interactions.

Interaction Type	FPPP	GGPW	IPHF	IPKF	IPRR	LPHF	LPRR
Hydrogen Bonding	3	5	8	4	11	6	7
Hydrophobic Interactions	6	4	10	7	4	7	5
Electrostatic Interactions	0	2	3	2	5	1	4

**Table 6 foods-13-02798-t006:** In silico tools for salty peptide screening.

List	Binding Energy (kcal/mol)	Toxicity	Sensitization	Stability
FPPP	−9.4	−	−	−
IPHF	−9.1	−	−	+
LPHF	−9.1	−	−	+
GGPW	−9	−	−	+
IPKF	−8.6	−	−	+
IPRR	−8.8	−	−	−
LPRR	−8.6	−	−	−

Note: “−” means that salty peptides were identified by in silico tools and have no such characteristics. “+” means that salty peptides were identified by in silico tools to have such characteristics.

## Data Availability

The original contributions presented in the study are included in the article, further inquiries can be directed to the corresponding authors.

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
