# Peer review of "Enzymatic Preparation, Identification by Transmembrane Channel-like 4 (TMC4) Protein, and Bioinformatics Analysis of New Salty Peptides from Soybean Protein Isolate"

_foods, 2024, doi:10.3390/foods13172798_

Round 1

Reviewer 1 Report

Comments and Suggestions for Authors

The manuscript presented by Zhao and co-workers, entitled „Enzymatic Preparation, Identification by Transmembrane Channel-like 4 (TMC4) Protein and Bioinformatics Analysis of New Salty Peptides from Soybean Protein Isolate” (foods-3153994) is an article presenting preparation of novel salty peptides through enzymatic hydrolysis, separation, purification, and identification from Soybean Protein Isolate. The presented manuscript is interesting, but some changes must be made.

Please find below my major comments. 

- abstract does not present the scientific importance of the study, 

- materials and methods section – not all of the used chemicals were presented in the first paragraph (HCl). Additional question is what was the purity of the used chemicals? 

- 2.7. Identification of the peptide sequence – what was the type of used LC-MS device (company, country?), determine the column parameters, determine the ESI source parameters. How were the MS/MS conditions optimized (isolation window, collision energy)? What was the m/z accuracy?

 - table 3 – is the m/z values are presented please determine the ion type ( [M+H]+, other)

- what was the sequence coverage determine by PEAks software?

- what were the PEAKs parameters used during the MS/MS data analysis?

- The presented literature does not fully present the latest scientific achievements in the presented topic and should be completed.  

Comments on the Quality of English Language

Moderate English editing is needed.

Author Response

Response to Reviewer Comments

1. Summary

Thank you very much for taking the time to review this manuscript (foods-3153994). The following is a detailed reply, with the corresponding changes marked in blue in the resubmitted document.At the same time, We have made appropriate modifications to the English expression of the full text.

2. Questions for General Evaluation

Reviewer’s Evaluation

Response and Revisions

Does the introduction provide sufficient background and include all relevant references?

Can be improved

Thank you for pointing this out. We have supplemented the background and references.

Is the research design appropriate?

Can be improved

Thank you for pointing this out. We've made some adjustments to the experimental design.

Are the methods adequately described?

Must be improved

Thank you for pointing this out. We've added to the methodology.

Are the results clearly presented?

Must be improved

Thank you for pointing this out. We have supplemented the findings.

Are the conclusions supported by the results?

Can be improved

We have checked the results and conclusions.

3. Point-by-point response to Comments and Suggestions for Authors

Comments 1: abstract does not present the scientific importance of the study, 

Response 1: Thank you for pointing this out. We agree with this comment. I have, accordingly, added the scientific importance to emphasize this point.You can see the changes in the Abstract.

Comments 2: materials and methods section – not all of the used chemicals were presented in the first paragraph (HCl). Additional question is what was the purity of the used chemicals?

Response 2: Agree. We have, accordingly, added used chemicals to emphatically explain this point (Paragraph 1, Line 2, Page 3).

Comments 3: 2.7. Identification of the peptide sequence –what was the type of used LC-MS device (company, country?), determine the column parameters, determine the ESI source parameters. How were the MS/MS conditions optimized (isolation window, collision energy)? What was the m/z accuracy?

Response 3: Thank you for pointing this out. This question is very important for the elaboration of the method section in the manuscript. We have, accordingly, added the information and parameters of the device to emphasize this point. You can see the changes in the Paragraph 4, Line 2, Page 4.

The ESI source parameters: Spray Voltage (+): 1900.00

Capillary Temperature (+ or +-): 320.00

Capillary Temperature (-): 250.00

Sheath Gas (+ or +-): 0.00

Sheath Gas (-): 0.00

Aux Gas (+ or +-): 0.00

Aux Gas (-): 0.00

Spare Gas (+ or +-): 0.00

Spare Gas (-): 0.00

Max Spray Current (+): 50.00

Max Spray Current (-): 50.00

Probe Heater Temp. (+ or +-): 350.00

S-Lens RF Level: 50.00

Because it is not DIA detection, there is no isolation window. Collision energy is 27.

Comments 4: table 3 – is the m/z values are presented please determine the ion type ( [M+H]+, other)

Response 4: Thank you for pointing this out. We agree with this comment. The ion type is [M+H]+.

Comments 5: what was the sequence coverage determine by PEAks software?

Response 5: Thank you for pointing this out. We have, accordingly, added the sequence identification additional table to emphasize this point. 

Comments 6: what were the PEAKs parameters used during the MS/MS data analysis?

Response 6: Thank you for pointing this out. The PEAKs parameters used Peak area during the MS/MS data analysis.

Comments 7: The presented literature does not fully present the latest scientific achievements in the presented topic and should be completed.

Response 7: Thank you for pointing this out. We have added to the literature on recent research advances on this topic.

Comments 7: The presented literature does not fully present the latest scientific achievements in the presented topic and should be completed.

Response 7: Thank you for pointing this out. We have added to the literature on recent research advances on this to

Reviewer 2 Report

Comments and Suggestions for Authors

This manuscript investigates the saltiness and bioinformatics of the soybean peptides from enzymatic hydrolysis. The manuscript claims to introduce novel peptides with higher salty taste than known salty peptides. The experiment is well designed and result are discussed adequately.  

There are few minor comments you can find in the attached file. 

I strongly recommend authors to provide all minor details. 

Author Response

Response to Reviewer Comments

1. Summary

Thank you very much for taking the time to review this manuscript (foods-3153994).The following is a detailed reply, with the corresponding changes marked in red in the resubmitted document.

2. Questions for General Evaluation

Reviewer’s Evaluation

Response and Revisions

Does the introduction provide sufficient background and include all relevant references?

Can be improved

Thank you for pointing this out. We have supplemented the background and references.

Is the research design appropriate?

Yes

Are the methods adequately described?

Can be improved

Thank you for pointing this out. We've added to the methodology.

Are the results clearly presented?

Can be improved

Thank you for pointing this out. We have supplemented the findings.

Are the conclusions supported by the results?

Yes

3. Point-by-point response to Comments and Suggestions for Authors

Comments 1: What is the composite hydrolysis?

Response 1: Thank you for pointing this out. We agree with this comment. Therefore, we have changed the complex hydrolysis to continuous enzymatic hydrolysis. (in the Abstract)

Comments 2: SPI hydrolysate

Response 2: Agree. We have, accordingly, changed SPI to SPI hydrolysate to emphasize this point. (in the Abstract)

Comments 3: Provide full form of LC-MS/MS

Response 3: Agree. We have, accordingly, changed LC-MS/MS to Liquid Chromatography-Mass Spectrometry/Mass Spectrometry to emphasize this point. You can see the changes in the Abstract.

Comments 4: SPI peptides?

Response 4: Thank you for pointing this out. After a series of steps such as enzymolysis, separation and purification, identification and molecular docking, SPI obtained salty peptides. So, we wrote SPI here.

Comments 5: As mention in title, are these novel peptides?

Response 5: Thank you for pointing this out. These 7 peptides (FPPP, GGPW, IPHF, IPKF, IPRR, LPRR, and LPHF) are theoretically salty peptides obtained through molecular docking in this study, and then basic bioinformatics analysis is used to screen out 4 peptides (IPHF, LPHF, GGPW, and IPKF) from the above 7 peptides, which are novel salty peptides referred to in the title.

Comments 6: Elaboration on physiological effects.

Response 6: Thank you for pointing this out. We agree with this comment. Therefore, we have added “it regulates the osmotic balance between cells and blood and normal water and salt metabolism”. You can see the changes in the Paragraph 1, Line 2, Page 1.

Comments 7: Provide brief background why is this challenging to food industries?

Response 7: Thank you for pointing this out. The previous article has explained the harm of excessive intake of salt to the human body, so we have changed this sentence into a more appropriate sentence. However, the addition of salt makes an important contribution to the flavor and edible quality of many foods (e.g., breads, processed soy products, potato chips, biscuits, canned products). In response to increasing consumer demand for healthier food options, the food industry will make more far-reaching efforts to reduce salt. You can see the changes in the Paragraph 1, Line 7, Page 1.

Comments 8: Full form of NaCl

Response 8: Agree. We have, accordingly, changed NaCl to sodium chloride. You can see the changes in the Paragraph 1, Line 6, Page 2.

Comments 9: As legumes are richer in globulin proteins, does the protein composition have any roles besides the protein content in the sources?

Response 9: Thank you for pointing this out. We agree with this comment. Therefore, we have added “and rich in salty and umami amino acids” in it. You can see the changes in the Paragraph 2, Line 4, Page 2.

Comments 10: Has the enzymatic hydrolysis of SPl to obtain salty peptides attempted before? Is this a novel technique leading to novel peptide production? Highlight the novelty of work

Response 10: Thank you for pointing this out. We agree with this comment. We have revised the original text to reflect the novelty of the research more. You can see the changes in the Paragraph 2, Line 9, Page 2.

Comments 11: Any research on soy protein based other salty peptides?

Response 11: Thank you for pointing this out. But We am sorry to say that the only research We have found is on salty peptides from soy products, but We have not found any research literature on salty peptides from soy protein isolate. Therefore, this is also one of the innovative points of my topic selection.

Comments 12: Hydrolysates? keep it uniform

Response 12: Agree. We have, accordingly, changed SPI digests to SPI enzymatic hydrolysate. You can see the changes in the Line 6, Page 3.

Comments 13: Which are?

Response 13: Thank you for pointing this out. We have, accordingly, changed internal and external enzymes to pepsin and flavourzyme. You can see the changes in the Paragraph 2, Line 2, Page 3.

Comments 14: Brief information on standard run to identify molecular weight. Or analytical tool used to identify molecular weight distribution

Response 14: Thank you for pointing this out. We agree with this comment. The relative correction of Mark-Houwink parameters can be performed by using narrow distribution PEO or Pullulan polysaccharide samples as standard curves

Comments 15: Hydrolysatses is it? How was the supernatant prepared.

Response 15: Thank you for pointing this out. 20mgSPIEH freeze-dried powder was dissolved in 10 mL~15 mL of 6mol/L hydrochloric acid solution, and the supernatant was obtained by centrifugation. This step is performed in accordance with GB 5009.124.

Comments 16: And how was amino acid content calculated. brief introduction

Response 16: Thank you for pointing this out. In this study, amino acid content was calculated by an amino acid analyzer (ninhydrin post-column derived ion exchange chromatograph). This step is calculated in terms of GB 5009.124.

Comments 17:  Hydrolysates?

Response 17: Thank you for pointing this out. I have, accordingly, changed lyophilized powders to lyophilized powders of enzymolysis. You can see the changes in the Paragraph 5, Line 5, Page 3.

Comments 18: Sample from both UF and GFC or GFC only, specify which samples peptide sequences were identified.

Response 18: Thank you for pointing this out. We have, accordingly, changed “The samples were” to “The sample from GFC was”. You can see the changes in the Paragraph 4, Line 1, Page 4.

Comments 19: Any statistical analysis? How many trial were conducted?

Response 19: Thank you for pointing this out. We have, accordingly, added “the experiment was repeated three times”. You can see the changes in the Paragraph 4, Line 2, Page 5.

Comments 20: Molecular weight distribution of the hydrolysate

Response 20: Thank you for pointing this out. We have, accordingly, changed “Molecular weight distribution analysis” to “Molecular weight distribution of the hydrolysate”. You can see the changes in the Paragraph 5, Page 5.

Comments 21: Start with your result first, this statement can be used to support your findings in the discussion. This sentence is repeated in discussion in other sections too.

Response 21: Thank you for pointing this out. We have, accordingly, reversed the sequence of paragraph statements. You can see the changes the Paragraph 5, Line 1, Page 5.

Comments 22: ls it for soy protein?

Response 22: Thank you for pointing this out. As mentioned above, we have not found any literature on soy protein salty peptides, and the salty peptides extracted from bovine bone cited in this paper are compared with the results of this paper.

Comments 23: why is it complex? reconsider in the use of specific terms

Response 23: Thank you for pointing this out. We have, accordingly, changed “the dual-enzyme complex hydrolysis” to “the dual-enzyme continuous enzymatic hydrolysis”. You can see the changes in the Paragraph 5, Line 8, Page 5.

Comments 24: Is it 100 kDa recheck,or arrange in an decreasing order. (Table 1)

Response 24: Thank you for pointing this out. We have, accordingly, changed “100 kDa” to “100 Da”. You can see the changes in the Table 1.

Comments 25: Is the amino acid profile of SPIEH unique to SPI? Any limiting amino acid? Significance of calculating amino acids of SPIEH? any nutritional benefits?

Response 25: Thank you for pointing this out. Amino acid content was measured using SPIEH as raw material, so the amino acid profile of SPIEH is unique to SPI. Met is limiting amino acid. The purpose of amino acid content detection is to preliminarily determine the possible presence of salty peptides in SPIEH according to the amino acid types and contents. And contains a large number of essential amino acids, with nutritional value.

Comments 26: Is this column necessary content expressed in (%) g/100 g (Table 2)

Response 26 Thank you for pointing this out. I have, accordingly, deleted this column from Table 2. You can see the changes in the Table 2.

Comments 27: In the method section before purification of peptides, it is mentioned sensory evaluation wasdone. Contrary, here you have mentioned that UF separated peptides were analysed for sensory Please keep the result and method section in aligment to avoid confusion.

Response 27: Thank you for pointing this out. We have refined and emphasized the salt taste evaluation system. You can see the changes in the Paragraph 1, Line 2, Page 7.

Comments 28: Justification why smaller peptides could have more salty taste.

Response 28: Thank you for pointing this out. We have already added explanations in the text. The saltiness of a peptide usually refers to the electrolyte content or charge effect of the peptide in solution. The reason why smaller molecular weight peptides may exhibit higher saltiness may be because smaller molecular weight peptides have an increased charge density, and their stronger water solubility makes them easier to dissociate. At the same time, the higher diffusion rate makes it easier for their charged groups to exchange with ions in solution, thus increasing the salinity. You can see the changes in the Paragraph 1, Line 13, Page 7.

Comments 29: Keep similar alignment in the method section.

Response 29: Thank you for pointing this out. We have checked the correspondence between the method and the result, and now it is consistent. You can see the changes in the Paragraph 1, Line 19, Page 7.

Comments 30: It is clear only now that purified peptides were further evaluated for sensory values can you make this clear in method to avoid confusion. Also What is the variation in G1 to G4, is it molecular size? or other factors

Response 30: Thank you for pointing this out. The ambiguity of the previous sensory evaluation has been resolved. Then, the difference between different peaks is that the peak time varies with different molecular weight.

Comments 31: What is the correlation between sensory and electronic tongue score? (Fig. 1B)

Response 31: Thank you for pointing this out. The score of sensory evaluation is directly proportional to that of electronic tongue.

Comments 32: What does AHS, CTS legends mean? (Fig. 1B)

Response 32: Thank you for pointing this out. We have, accordingly, added “Note: AHS means acid; CTS means salty; NMS means umami; SCS means bitter; ANS means sweet; CPS, CKS are chiasma type”. You can see the changes in the Fig. 1.

Comments 33: Image resolution needs to be improved. difficult to read legends and data (Fig. 1)

Response 33: Thank you for pointing this out. Due to file limitations, we initially uploaded a compressed package of high-definition pictures and will upload it again in the future.

Comments 34: Recheck this. It is confusing and looks like there is error in the text too. If D and E image are electronic tongue score of GPC fractions? Can sensory evaluation and electronic tongue score can be presented in the similar graph for...

Response 34: Thank you for pointing this out. We have checked and corrected the errors in the article. D is electronic tongue score of GFC fractions, E is Sensory evaluation score of GFC fractions. The electronic tongue score is more intuitive to express with radar map, and the sensory evaluation score is more intuitive with column chart, and it is convenient to distinguish the two salt taste evaluation methods. You can see the changes in the Fig. 1.

Comments 35: Start with result and sample information.

Response 35: Thank you for pointing this out. We have adjusted the word order in the original text to emphasize the logic of the narrative. You can see the changes in the Paragraph 1, Line 6, Page 8.

Comments 36: Theoritical information. How do you relate this information to you results below?

Response 36: Thank you for pointing this out. This sentence is written about the principle of LC-MS/MS, which may not be closely related to the narration of this paragraph, so We choose to delete it after careful consideration, so as to make the original text more complete and orderly.

Comments 37:  Can this information be provided in supplementary file?

Response 37: Thank you for pointing this out. We have provided supplementary documents to this section. You can see the changes in the supplementary documents.

Comments 38: why and how? chemistry behind this (Paragraph 2, Line 11, Page 8)

Response 38: Thank you for pointing this out. We have already added explanations in the text: The saltiness of a peptide usually refers to the electrolyte content or charge effect of the peptide in solution. The reason why smaller molecular weight peptides may exhibit higher saltiness may be because smaller molecular weight peptides have an increased charge density, and their stronger water solubility makes them easier to dissociate. At the same time, the higher diffusion rate makes it easier for their charged groups to exchange with ions in solution, thus increasing the salinity. You can see the changes in the Paragraph 1, Line 13, Page 7.

Comments 39: Can you elaborate this more. Amino acid types. Peptide chain length. What does the column m/z in table 3 represents

Response 39: Thank you for pointing this out. We have elaborated further on amino acid types and peptide lengths. You can see the changes in the Paragraph 1, Line 12, Page 8. In mass spectrometry, m/z is a very important parameter, which represents the mass-charge ratio. Each peak on the mass chart represents an ion whose m/z value is the ratio of its mass to its charge.

Comments 40: Peptide segments of G2. Mention the peculiarity of G2 than others

Response 40: Thank you for pointing this out. G2 peak has a higher saltiness than other peaks, so the freeze-dried samples of G2 peak were sequenced.

Comments 41: Similar brief intro on results section are expected for others too. This provide information on experiment and result premise and help to understand what to expect in this section.

Response 41: Thanks for your affirmation. We have made corresponding modifications elsewhere in the paper, such as explaining why the shorter the peptide, the saltier the peptide. You can see the changes in the Paragraph 1, Line 13, Page 7.

Comments 42: Full form and why this is important to be the highest.

Response 42: Thank you for pointing this out. We have, accordingly, changed “pLDDT (78.3) and the highest PTM score (0.773)” to “the predicted local distance difference test (pLDDT) (78.3) and predicted TM-score (PTM) (0.773)”. You can see the changes in the Paragraph 2, Line 5, Page 14. pLDDT was used to assess the local prediction accuracy of each residue. The higher the PLDDT value, the more confident the prediction model in this region and the higher the accuracy of its structure prediction. PLDDT usually takes on a range of 0 to 100. PTM was used to assess the accuracy of global topological predictions for the entire protein. The higher the PTM, the more accurate the model's prediction of protein folding as a whole. Together, they measure the predictive quality of the model. A high PLDDT means that the model's predictions for a specific region are accurate, and a high PTM means that the overall structure is predicted correctly. Both of these indicators are as high as possible to ensure that the predicted protein structure is both accurate in local detail and correct as a whole.

Comments 43: need a brief intro

Response 43: Thank you for pointing this out. We have adjusted the word order and content of this paragraph to make it more complete and orderly. You can see the changes in the Paragraph 1, Line 1, Page 12.

Comments 44: Supplementary information. Energy binding for other peptides besides the selected 7 ones.

Response 44: Thank you for pointing this out. We have provided supplementary documents to this section. You can see the changes in the supplementary documents.

Comments 45: Similar justification is needed for other result result section.

Response 45: Thanks for the reviewer's comments on me. We have modified other similar places, such as explaining why the shorter the peptide, the saltier the peptide. You can see the changes in the Paragraph 1, Line 13, Page 7.

Comments 46: What is the extent of difference?

Response 46: Thank you for pointing this out. We have supplemented the sequence of control group peptides, and you can see the difference between them in the comparison of binding energy size above. You can see the changes in the Paragraph 1, Line 18, Page 12.

Comments 47: Control peptides?

Response 47: Thank you for pointing this out. We have, accordingly, changed “known salty peptides” to “control peptides”. You can see the changes in the Paragraph 1, Line 6, Page 16.

Comments 48: This is well written. keep in mind to align method and result to similar manner.

Response 48: Thank you for your affirmation of me, we have made some corresponding modifications in the article.

Comments 49: Mention them

Response 49: Thank you for pointing this out. We have, accordingly, changed “double enzyme complex” to “pepsin and flavourzyme”. You can see the changes in the Paragraph 3, Line 2, Page 17.

Comments 50: hydrolysates

Response 50:  Thank you for pointing this out. We have, accordingly, changed “protein” to “hydrolysates”. You can see the changes in the Paragraph 3, Line 2, Page 17.

Round 2

Reviewer 1 Report

Comments and Suggestions for Authors

The revised version of the manuscript presented by Zhao and co-workers meets my requirements. The authors made corrections, which significantly improved the manuscript. The authors presented comments and answerers to all my questions and doubts. 

Comments on the Quality of English Language

Minor English editting is needed.